# Nonalcoholic Fatty Liver Disease (NAFLD) and Hepatic Cytochrome P450 (CYP) Enzymes

**DOI:** 10.3390/ph13090222

**Published:** 2020-08-29

**Authors:** Rohitash Jamwal, Benjamin J. Barlock

**Affiliations:** Biomedical and Pharmaceutical Sciences, University of Rhode Island, Kingston, RI 02881, USA; bbarlock@my.uri.edu

**Keywords:** NAFLD, NASH, drug metabolism, CYP450, LC–MS/MS, proteomics, PBPK

## Abstract

Nonalcoholic fatty liver disease (NAFLD) is characterized by excessive fat in the liver. An international consensus panel has recently proposed to rename the disease to metabolic dysfunction associated with fatty liver disease (MAFLD). The disease can range from simple steatosis (fat accumulation) to nonalcoholic steatohepatitis (NASH) which represents a severe form of NAFLD and is accompanied by inflammation, fibrosis, and hepatocyte damage in addition to significant steatosis. This review collates current knowledge of changes in human hepatic cytochrome P450 enzymes in NAFLD. While the expression of these enzymes is well studied in healthy volunteers, our understanding of the alterations of these proteins in NAFLD is limited. Much of the existing knowledge on the subject is derived from preclinical studies, and clinical translation of these findings is poor. Wherever available, the effect of NAFLD on these proteins in humans is debatable and currently lacks a consensus among different reports. Protein expression is an important in vitro physiological parameter controlling the pharmacokinetics of drugs and the last decade has seen a rise in the accurate estimation of these proteins for use with physiologically based pharmacokinetic (PBPK) modeling to predict drug pharmacokinetics in special populations. The application of label-free, mass spectrometry-based quantitative proteomics as a promising tool to study NAFLD-associated changes has also been discussed.

## 1. Introduction

Substantial changes in the dietary habits of our generation are fueling an epidemic of various metabolic disorders. Nonalcoholic fatty acid liver disease (NAFLD) is one such metabolic syndrome which is rising at an alarming rate [1,2]. An international consensus panel has recently suggested using metabolic dysfunction associated with fatty liver disease (MAFLD) instead of NAFLD to more accurately reflect the heterogeneity of the disease [3]. Nevertheless, for the sake of this review, we will continue the use of NAFLD, which has been widely accepted in the literature. The prevalence of NAFLD is higher in patients with diabetes, obesity, hyperlipemia, hypertension, and hypertriglyceridemia [4,5]. NAFLD is characterized by the presence of greater than 5% of hepatic fat in people without significant alcohol intake (<20 g per day for women, <30 g per day for men) [6,7]. Fat accumulation in hepatocytes is triggered by various mechanisms, which include the increased hepatic uptake of circulating fatty acids, increased hepatic de novo fatty acid synthesis, decreased hepatic beta oxidation, and decreased hepatic lipid export [8]. The disease is characterized by the accumulation of free fatty acids and triglycerides in the hepatocytes, and severity ranges from benign steatosis to nonalcoholic steatohepatitis *(*Figure 1*)* [9,10]. NASH, characterized by significant lobular inflammation, hepatic fibrosis, and hepatocyte necrosis, can progress to life-threatening liver cirrhosis and hepatocellular carcinoma (HCC) [11].

## 2. Pathogenesis, Epidemiology and Risk Factors

The current “multiple-hit” theory for the pathogenesis of NAFLD proposes the first hit as the accumulation of lipid droplets (triglycerides) in >5% of hepatocytes [12]. A successive second hit is characterized by excessive free radical and pro-inflammatory cytokine formation, leading to inflammation, necrosis, and, consequently, fibrosis [12]. Several histologic scoring systems have been introduced in the past decade for the diagnostic evaluation of different stages of NAFLD [13,14,15]. Pathologists commonly differentiate stages of NAFLD using a semi-quantitative evaluation based on steatosis, lobular inflammation, hepatocellular ballooning, and/or fibrosis [16]. Liver biopsies are the gold standard for the diagnosis of NAFLD but ethical, and feasibility constraints limit the direct assessment of its prevalence. The global prevalence of NAFLD diagnosed by imaging was estimated to be 25.2%, and was >30% in the Middle East and South America [17]. The lowest prevalence of NAFLD was reported in Africa (13.5%). The prevalence of NAFLD diagnosed with ultrasonography was 24.13%. The overall prevalence of nonalcoholic steatohepatitis (NASH) in general populations is estimated to be between 1.5 and 6.5% [17]. In the USA, NAFLD is also associated with significant economic (~$103 billion) and clinical burden (~64 million people projected to have NAFLD) [18]. Ethnic differences exist in the prevalence of NALFD and Hispanics are more susceptible to the disease compared to Caucasians and African Americans [19]. The disease was initially thought to be present only in obese adults; however, recent studies have shown its prevalence in people with a healthy BMI as well as children [20,21,22].

Certain genetic factors are also responsible for the predisposition to NAFLD. PNPLA3 (Patatin-like phospholipase domain-containing protein 3) is a multifunctional enzyme involved in the hydrolysis of triacylglycerol (TAG) in the liver, and the rs738409 variant is the strongest genetic risk factor for NAFLD [23]. Recent genome-wide association studies (GWAS) have identified increased NAFLD susceptibility in variants of *TM6SF2* (rs58542926; transmembrane 6 superfamily member 2), *GCKR* (rs780094, Glucokinase regulator), *NCAN* (rs2228603, Neurocan), and *LYPLAL1* (rs12137855, Lysophospholipase-like 1) [24,25]. The association of an rs641738 variant of *MBOAT7-TMC4* locus (membrane-bound O-acyltransferase domain-containing 7, transmembrane channel-like 4) is controversial and inconclusive [26,27].

Physical findings show that patients with NAFLD often have obesity and hepatomegaly (enlarged liver) due to fat infiltration of the liver. NAFLD patients show mild or moderate elevations in the aspartate aminotransferase (AST) and alanine aminotransferase (ALT), although normal aminotransferase levels do not exclude NAFLD. When elevated, the AST and ALT are typically 2–5-fold the upper limit of normal, with an AST to ALT ratio of less than one (>1.5 for alcoholic liver disease) [28]. Alkaline phosphatase may also be elevated 2–3-fold the upper limit of normal and patients may have an elevated serum ferritin concentration or transferrin saturation. Decreased hepatic attenuation on computed tomography (CT) and an increased fat signal on magnetic resonance imaging (MRI) is generally evident in radiographic findings [29].

## 3. Classification Systems for NAFLD

NAFLD is a complex disease and differentiating definite non-alcoholic steatohepatitis (NASH) from non-alcoholic fatty liver (NAFL) can be equally complicated for basic researchers as for diagnostic pathologists. The gray zone of the distinction between NAFL and NASH is precarious, and diagnosis often varies dramatically among pathologists due to the heterogenous histopathologic spectrum of NAFLD and its progression over time [30]. Controversy exists over the use of these classifications which hass led to the misuse of scoring systems [13,31,32]. The categorical nature of the histopathological scoring system adds to the discrepancies in the diagnosis of the type of NAFLD. Since the categories are not well defined, eventually there are varying interpretations and conclusions from researchers and pathologists [16]. The additive nature of recent scoring systems (NAS or SAF) deconvolutes the contribution of each histologic legion.

Interestingly, the NAS scoring system was not designed to be used as a diagnostic tool for the determination of NASH versus NAFL [13,32]. It was intended to evaluate the changes in histological lesions (steatosis, inflammation, hepatocyte ballooning, fibrosis) that can occur over time [31]. Fatty liver inhibition of progression (FLIP) algorithm was proposed to improve the consistency in the diagnosis of NASH in adults and takes into account fibrosis along with steatosis and activity score (inflammation, ballooning) [30]. The use of different classification systems may, therefore, contribute to significant variability seen in the literature on the subject. Surgical hepatitis and potential differences between the biopsy site (right or left lobe) also confound the diagnosis of histologic lesions and subsequently the proper classification of disease [16].

## 4. Treatment Strategies to Manage NAFLD

Hepatic pathological conditions have been known to impact the abundance of DMEs and transporters in the liver, leading to altered drug profiles and often to side effects [33]. Despite the widespread prevalence of NAFLD/NASH, currently, there are no pharmacological therapies available for its treatment and it involves the management of associated conditions including obesity, diabetes, hyperlipidemia [34,35,36]. The current standard of care in the absence of FDA-approved drugs is through lifestyle and dietary interventions, weight loss, exercise, and management of comorbidities [7]. Weight loss is usually the first and most common intervention recommended for any metabolic syndrome [37]. Similarly, lifestyle modifications through exercise and dietary restriction are considered vital in the management of NAFLD [38]. Patients with the disease are advised to avoid intake of positive calorie foods such as soda and sweetened drink which are rich in simple carbohydrates that are readily absorbed [39]. Patients are also asked/recommended to avoid diets rich in cholesterol, fructose, and other saturated fats which are often linked with the progression of NAFLD [40,41].

Interestingly, consumption of fructose-sweetened and nonglucose-sweetened beverages has been associated with elevated insulin resistance and an increase in visceral adiposity and lipids in overweight and obese humans [42]. Ryan et al. found that the Mediterranean diet improved insulin sensitivity and reduced hepatic steatosis in NAFLD patients with insulin resistance [43]. A considerable effort is currently underway in the development of therapeutic agents for the treatment of NAFLD, and multiple molecular pathways are now being targeted for drug development [44]. However, a lack of clear unifying pathogenic mechanism and multifactorial nature of the disease has made it difficult to prioritize druggable targets for NAFLD. This has been reflected by the multitude of pharmacological targets currently pursued by the pharmaceutical industry. New therapies in development target one or more of the following pathways: (a) hepatic fat accumulation and insulin resistance; (b) oxidative stress, inflammation and apoptosis; and (c) hepatic fibrosis [45]. An overview of some of the therapeutic agents in clinical development is given in Appendix A. Despite the widespread prevalence of disease and considerable drug development efforts, questions still outnumber the answers for drugmakers. Given the poor performance of highly anticipated drug candidates from Gilead and GenFit in phase III, it has become clear in recent years that the conventional mono-target approach may not be able to achieve the desired primary endpoints in clinical studies. Leoni et al. have reviewed current guidelines in the diagnosis of NAFLD as well as the areas of therapeutic focus [7].

## 5. Drug Metabolism Enzymes and NAFLD

The human liver, facilitated by several drug-metabolizing enzymes (DMEs), is the primary organ responsible for the elimination of xenobiotics and endogenous compounds. DMEs are responsible for the metabolism of several chemicals which include xenobiotics such as drugs, pesticides, and endogenous substrates such as steroids and bile acids [46]. DMEs are broadly classified into phase-I enzymes, which are mostly oxidative, reductive or hydrolytic, and phase-II enzymes, which are conjugative. Major proteins involved in oxidative biotransformation are cytochrome P450 (CYP450s), flavin-monooxygenases (FMOs), monoamine oxidases (MAOs), alcohol and aldehyde dehydrogenases, and aldehyde and xanthine oxidase [47]. Aldo-keto reductases (AKRs), azo- and nitro-reductases constitute reductive enzymes involved in metabolism, whereas epoxide hydrolases, esterases, and peptidases are responsible for the bulk of hydrolysis reactions in the liver [47]. Cytochrome P450 superfamily enzymes are involved in the majority of reductive reactions and are reported to be responsible for the metabolism of ~70–80% of all the available drugs on the market [46]. Conjugation reactions in the liver are carried out by uridine diphosphate-glucuronosyltransferases (UGTs), sulfotransferases (SULTs), N-acetyltransferases (NATs), glutathione S-transferases (GSTs), amino acid conjugation enzymes and methyltransferases [47,48]. DMEs in the liver control systemic levels of drugs and drive the efficacy as well as adverse events in the body. A change in liver physiology and protein abundance of drug-metabolizing enzymes in a disease state may compromise the efficacy of drugs and/or lead to adverse effects. Metabolic disorders such as NAFLD are likely to influence several mechanisms governing drug metabolism including changes in the activity of enzymes that break down drug molecules. This may necessitate the need for clinical dose optimization in patients with such disorders. The majority of studies evaluating the effect of NAFLD on drug disposition proteins draw their conclusions from studies performed in preclinical animal models. A careful interrogation often suggests a complex and heterogeneous alteration in drug-metabolizing enzymes and transporters in humans. Such discrepancies are partly due to species differences and lack of animal models which accurately reflect the complexity and pathophysiology of human disease.

## 6. Preclinical Studies

Preclinical species have played a key role in our understanding of NAFLD [49]. Genetic as well as dietary animal models have been developed to understand the disease. Common genetic mice models of NAFLD include leptin-deficient (*ob*/*ob*), leptin receptor-deficient (*db*/*db*), and low-density lipoprotein-deficient mice. Even though low leptin levels are not observed in human NAFLD, leptin-deficient mice (*ob*/*ob*) exhibit obesity, hyperlipidemia, insulin resistance, and steatosis but without fibrosis [50,51].

However, fibrosis can be induced in leptin receptor-deficient mice (*db*/*db*) from external stimuli. Low-density lipoprotein receptor (LDLR)-deficient mice exhibit a pathology similar to *db/db* mice and can develop fibrosis. Not all studies in mice models are consistent among different models; the results have shown to vary according to the diet and species [52,53,54]. Dietary models are developed using methionine and choline-deficient (MCD) and hypercaloric diets [55]. MCD models exhibit steatosis, inflammation, and fibrosis similar to human disease but significantly differ in the metabolic phenotype of the disease. MCD-fed mice show increased insulin sensitivity, significant weight loss, and low blood glucose.

Similarly, the hypercaloric diet (Western-like diet) model show steatosis, inflammation, fibrosis but take a significant time for disease induction. DIAMOND (a diet-induced animal model of alcoholic fatty liver disease) and STAM (Stelic Animal Model) models are proposed to exhibit considerable similarity with human NAFLD in pathology and phenotype [56,57]. Both models gain weight and develop insulin resistance, steatosis, fibrosis, and hepatocellular carcinoma.

Another Western diet-induced (high fat, high fructose, and high cholesterol) murine NASH model in C57BL/6J mice was established to mimic the rapid progression of fibrosis and HCC in human [58]. Pro and cons of different animal models of NAFLD rodent models and their association with human NASH have been extensively reviewed elsewhere [59,60]. DME expression in various in vitro models for NASH has also been studied and was recently reviewed by Muller and Sturla [61].

Given the complexity of the human disease, no single animal model to date fully recapitulates the human disease state [59]. The failure of some compounds that showed a significant promise in preclinical studies has raised a concern about the inadequacies of animal models for the disease. ASP9831, a potent PDE4 inhibitor was being developed by Astellas Pharma to modulate cyclic adenosine monophosphate activity. Compared to placebo, ASP9831 failed to improve the biochemical parameters associated with NASH in a 12 week phase-II clinical trial [62]. Similarly, resveratrol was unable to improve hepatic steatosis and insulin sensitivity at pharmacological doses in an 8-week study [63].

## 7. Studies in Human

Disease-mediated changes have been known to impact the abundance of drug disposition proteins in the liver, hence leading to altered drug profiles [33,64,65,66]. Theoretically, an alteration in drug disposition proteins could lead to undesirable pharmacokinetic and pharmacodynamic outcomes. The systemic exposure of a substrate may increase if the CYP450 responsible for its metabolism and clearance is reduced. The opposite is true when the expression is increased.

Clinical studies in NAFLD are currently limited but are critical to understanding the implication of altered drug-metabolizing enzyme profile on the therapeutic result in the disease state. Current literature on studies with human tissue is confounding, with reports of an increase, decrease or nonsignificant change in the activity, protein, and/or mRNA levels of clinically relevant hepatic cytochrome P450 enzymes (Table 1). Genome-wide studies (GWAS) in NAFLD patients found no significant changes in DME proteins between normal and steatotic livers at mRNA expression level [67,68]. However, mRNA-based studies do not account for the potential contribution of post-transcriptional and post-translational changes relevant to protein expression or enzyme activity. A comprehensive table of alteration in drug metabolism enzyme expression or activity is given in Table 2. In general, a decrease in CYP3A4 and CYP2E1 activity appears to be dominant in the studies in subjects and human tissue [10,66,69]. Even though the effect of NAFLD on CYP450s has been extensively studied in vitro, our understanding of the disease-associated impact on other DMEs is limited due to the scarcity of studies. Similar to CYP450s, differential regulation appears to be at play for other DMEs including UGTs and SULTs in human NAFLD. Studies in human tissue found a minimal effect of NAFLD on UGT enzymes while a significant alteration was reported in hepatic sulfotransferase expression and activity [70]. NASH-mediated upregulation of UGT1A9, 2B10, and 3A1 mRNA was reported in the human liver [70]. It was also noteworthy that the protein expression of UGT1A9 and 1A6 decreased in NASH [70]. Elevated SULT1C4 mRNA was seen in NASH, whereas SULT1A1 and 2A1 protein levels were lower in disease samples compared to control samples [70].

## 8. Physiologically Based Pharmacokinetic (PBPK) Model

Physiologically based pharmacokinetic (PBPK) modeling has seen a rise in the last decade to support model-informed drug development [88]. PBPK modeling is a bottom–up simulation approach that takes into account multiple parameters specific to the drug, physiology of the species (different organs represented as compartments), and an understanding of the pharmacokinetic properties of the drug of interest [89]. Applications of PBPK models ranges from the prediction of human pharmacokinetics during first-in-human, drug–drug interaction as well as dosing recommendation in a special population.

System- or population-specific physiological parameters that are essential for the prediction of exposure include enzyme abundance, hepatic blood flow, and glomerular filtration rate. Drug-dependent parameters required for model development include molecular weight, solubility, permeability, blood-to-plasma ratio, and enzyme kinetics. An exhaustive list of different parameters required for building a PBPK model is discussed elsewhere [89]. Therefore, in vitro and in vivo parameters that accurately reflect the human disease are vital to predict and simulate in silico drug exposure. Some commercial PBPK platforms such as GastroPlus (Simulation Plus Inc.), Simcyp (Certara L.P.) PKSIM (Bayer), and CloePK (Evotec A.G.) are currently available.

Currently, there are insufficient data on the abundance of metabolic enzymes in NAFLD and there has been a limitation in the development of PBPK models. While a significant amount of literature is available for the levels of these proteins in healthy people, little is known about how NAFLD changes the concentration of these enzymes in the human liver. The information is essential to determine the influence of the disease on the drug disposition, but clinical studies in NAFLD subjects are limited. The use of physiologically based pharmacokinetic models for prediction of pharmacokinetics and drug metabolism in populations that present clinical challenges is increasing in popularity [90]. Vildhede et al. recently showed the promise of quantitative proteomics and a PBPK model for the prediction of morphine pharmacokinetics in subjects with NAFLD [91].

Hepatic blood flow is another critical parameter determining the rate of presentation of the drug for its metabolism in the liver. The hepatic portal vein (HPV) supplies 70%, whereas the hepatic artery (HA) is responsible for ~30% of the blood reaching the liver. The HPV supplies liver with nutrients and xenobiotics absorbed in the GI tract and the HA is responsible for carrying oxygen. Fat accumulation in hepatocytes was found to correlate with decreased HPV blood flow in NAFLD patients [92]. Hepatocyte ballooning associated with NASH causes sinusoidal distortion, leading to reduced intrasinusoidal volume and microvascular blood flow [93]. Impaired systemic circulation and modification of cellular membrane may also interrupt oxygen availability in NAFLD, leading to hypoxia and accelerated lipid droplet formation [94]. Blood flow change in early fibrosis has been attributed to outflow blockage in the liver sinusoidal area [95]. The changes in hepatic blood flow during different stages of NAFLD are given in Table 3 [96].

Additionally, enzyme distribution in mammalian liver exhibits metabolic zonation [97]. The expression of CYP450s and UGTs decreases with an increase in oxygen gradient, with higher expression in the perivenous region than periportal [98,99]. Available evidence suggests that CYP450s also exhibit a preferential induction in hepatocytes derived from the perivenous region of the liver [99]. Most studies lack in providing any information on the region from which the liver tissue is harvested and used for evaluation of protein expression. Given the heterogeneous distribution of CYP450s in the liver, the wide variability observed in the published studies with the human liver may be partially explained by sampling from different regions of tissue. Therefore, whenever possible, the information on the sampling region of the tissue used for a protein expression should be included with results.

**Table 3 pharmaceuticals-13-00222-t003:** Changes in hepatic blood flow in different stages of nonalcoholic fatty liver disease. Liver blood flow in the disease state was studied using xenon computed tomography. THBF: total hepatic blood flow; PVBF: portal vein blood flow; HABF: hepatic artery blood flow.

mL/min/100 g	Normal	NAFL	Early NASH	Advanced NASH
THBF	94.0 ± 17.8	66.4 ± 10.6	52.4 ± 12.3	47.1 ± 13.2
PVBF	72.4 ± 16.2	41.6 ± 5.6	33.6 ± 7.0	28.4 ± 6.3
HABF	21.2 ± 7.4	24.8 ± 8.5	18.7 ± 7.0	18.7 ± 8.4
Reference	[100]	[96]		

## 9. Label-Free Quantitative Proteomics

Traditionally, the level of expression is determined using Western blotting but the last decade has seen a rise in mass spectrometry-based methods for quantification of proteins. Omics technologies have been used extensively in the quest to identify novel biomarkers for NAFLD [101]. Targeted proteomics (SRSM, MRM) based approaches have also proven to be useful in quantification of DDPs. While the targeted quantification represents the most robust method of choice for absolute quantification, cost and significant time for optimization of mass spectrometer conditions for each targeted peptide limit its application to a few target proteins.

In contrast, label-free quantification (LFQ) has emerged recently as an alternative approach for comparative analysis of protein expression across different samples owing to the fast and low-cost of this technique [102]. LFQ approaches are relatively inexpensive compared to targeted MRM methods as there is no need to synthesize unique peptides for each protein and isotopically labeled isoforms of this peptide as the internal standard. Accurate and robust quantification with LFQ approaches is intricate, and different strategies for extracting quantitative data from LFQ analysis have been developed [102]. A comprehensive cost comparison of various mass spectrometry-based techniques reported significant cost savings with label-free-based quantitative proteomics [103].

Studies have shown that protein expression is a better surrogate than mRNA for the prediction of functional activity of cytochrome P450 enzymes [104]. Data-dependent (DDA) and data-independent analysis (DIA) are two common data collection modes in shotgun proteomics. In the DDA mode, the most abundant ion species from a precursor scan (MS1) at a given retention time are selectively selected for fragmentation (MS/MS). Alternatively, the precursors are selected in a specific m/z range and are fragmented without any prioritization to their relative abundance. This approach offers a more comprehensive and complete analysis of samples than traditional DDA. Sequential window acquisition of all theoretical mass spectra (SWATH–MS) is one such DIA technique that provides an alternative to DDA and targeted approaches for protein estimation [105]. As mentioned previously, SWATH is a DIA technique in which all the precursors within a predefined m/z are fragmented, and product ions of these precursors are recorded as a digital repository [106]. However, coeluting precursors and fragments at any given RT in DIA data make it difficult to select the correct peak without a robust spectral library. Therefore, a reference spectral library is often used for DIA and data is further deconvoluted and extracted using software such as OpenSWATH, SWATH 2.0, Spectronaut and Skyline [107]. A significant advantage of SWATH–MS over the other mass spectrometry methods is related to the ability to perform retrospective mining of the data. The targeted protein extraction can be improved by expanding the coverage of the reference spectral library and re-mining the DIA data. For instance, if the researcher comes up with a new hypothesis in the future, SWATH–MS data would allow interrogation of the existing data for additional protein/s of interest without the need for sample digestion or data reacquisition (Figure 2). Such a strategy offers a tremendous benefit concerning the saving of sample, time, and money.

Absolute protein concentrations are vital to simulate drug exposure using physiologically based pharmacokinetic (PBPK) models. However, despite all the advantages, DIA approaches are relative, and hence absolute protein concentration levels cannot be determined using techniques such as SWATH–MS. Alternatively, a spike in standard or targeted approach for a protein of interest would be needed to determine the protein levels. Global proteomics using DDA data and “total protein approach (TPA)” can instead be used to estimate protein concentrations. TPA is widely accepted and delivers protein concentrations without the need for the isotope-labeled spike in reference peptides [108]. TPA assumes that a protein’s abundance with a cell as a fraction of total protein is approximately the same as the proportion of its MS signal to the total MS signal of the cell. The absolute protein concentration (p), expressed as mol/g of total protein can further be calculated as described previously [108].

## 10. Concluding Remarks

The pandemic of NAFLD (or MAFLD) is upon us, and a widespread effort is currently underway to address different aspects of this multifaceted and complex metabolic disease. There is a lack of predictive biomarkers for NAFLD to delineate NASH from NAFL in human and mass spectrometry-based proteomics may hold promise to fill this gap. A lack of good preclinical models to recapitulate the spectrum of the disease also remains a significant challenge and care should be taken when extrapolating results from preclinical species to humans. The risk of alterations in the drug disposition proteins remains high in NAFLD due to significant structural and pathophysiological changes in the liver, the primary organ for drug disposition. PBPK models have been used recently to simulate the exposure of various drugs in special populations. However, most of our understanding of these models comes from research performed in healthy individuals, but little is known about the physiological changes of CYP450s in NAFLD. Moreover, after drug administration, there is also a need to understand how drug disposition proceeds in these disease/target populations. Two of the critical parameters governing the exposure include enzyme kinetics and the expression of proteins involved in the disposition of the compound. In summary, it is of paramount importance to determine how the disease alters the expression of the proteins involved in the disposition of the drugs. Additionally, to improve translation ability and accuracy of simulation models, it is essential to understand how the pharmacokinetics and other physiological parameters change in NAFLD.

## Figures and Tables

**Figure 1 pharmaceuticals-13-00222-f001:**
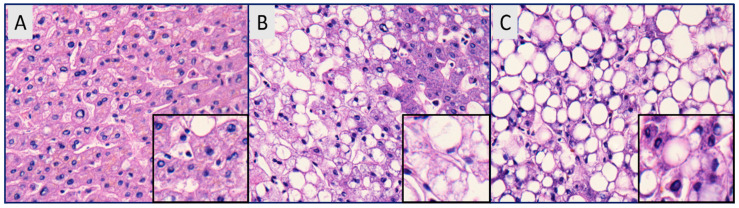
Hematoxylin and eosin staining of (**A**) normal liver with uniform chord-like arrangements of hepatocytes, (**B**) nonalcoholic fatty liver (NAFL) with lipid droplets, and (**C**) nonalcoholic steatohepatitis (NASH) with significant hepatic steatosis and infiltration of lymphomononuclear inflammatory cells [10].

**Figure 2 pharmaceuticals-13-00222-f002:**
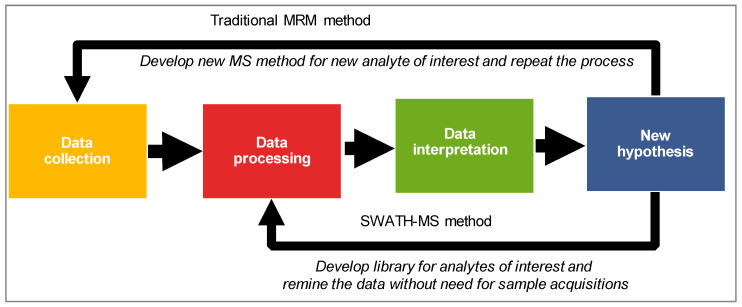
A graphical illustration of the advantage of SWATH–MS over the traditional MRM-based quantification method. MRM, multiple reaction monitoring; SWATH–MS, sequential windowed analysis of all the theoretical mass spectra.

**Table 1 pharmaceuticals-13-00222-t001:** Changes in expression and activity levels of major cytochrome P450 enzymes in humans. CYP, cytochrome P450; NASH: nonalcoholic steatohepatitis.

Enzyme	Disease Stage	mRNA Expression	Protein Expression	Activity	Ref.
CYP2E1	NAFLD	↑			[71]
	NAFLD			↔	[72]
	NAFLD			↓	[73]
	NAFLD progression	↓	↓	↔	[74]
	NASH	↑			[75]
	NASH			↑	[76]
	NASH			↑	[77]
	NASH			↑	[77]
	NASH		↑		[78]
	NASH		↑		[79]
	NAFL			↔	[80]
	NAFL	↑			[81]
CYP3A4/5	NASH	↔	↓	↓	[10]
	NAFL	↔	↓	↓	[10]
	NAFLD progression	↔	↔	↔	[74]
	NAFLD		↔		[82]
	Hepatocytes			↔	[80]
	Steatosis			↔	[83]
	NASH		↑		[79]
	NASH		↓		[78]
	NAFL	↔	↔	↓	[84]
CYP1A2	NAFLD progression	↔	↓	↓	[74]
	NAFL			↓	[80]
	NAFL	↓			[67]
CYP2A6	NAFLD progression	↑	↑	↑	[74]
	NASH	↓			[85]
	NASH		↑		[79]
CYP2B6	NAFLD progression	↑	↔	↔	[74]
	NASH	↓			[86]
	NASH	↓			[87]
CYP2C8	NAFLD progression	↔	↔	↔	[74]
CYP2C9	NAFLD progression	↔	↔	↑	[74]
CYP2C19	NAFLD progression	↔	↓	↓	[74]

**Table 2 pharmaceuticals-13-00222-t002:** Drug metabolism enzymes localization and commonly catalyzed biotransformation reaction in human liver [47].

Drug Metabolism Enzyme	Major Enzyme Isoform	Localization in Human Liver	Common Biotransformation Reaction
Cytochrome P450	CYP1A2, CYP2B6, CYP2C8, CYP2C9, CYPC19, CYP2D6, CYP3A4/5	Microsomes	Oxidation
Uridine 5′-diphospho glucuronosyltransferase	UGT1A1, UGT1A3, UGT1A4, UGT1A6, UGT1A9, UGT2B7, UGT2B10, UGT2B15	Microsomes	Conjugation
Sulfotransferases	SULT1A1, SULT1A3, SULT1E1, SULT2A1, SULT1B1	Cytosol	Conjugation
Aldo-keto reductases	AKR1A1, AKR1B1, AKR1C1, AKR1D1	Cytosol, microsome	Reduction
Flavin-containing monooxygenases	FMO3, FMO4, FMO5	Microsomes	Oxidation
Monoamine oxidase	MAO-A, MOA-B	Cytosol	Oxidation
N-acetyltransferases	NAT1, NAT2	Cytosol, mitochondria	Conjugation

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
