# Peer review of "Nonalcoholic Fatty Liver Disease (NAFLD) and Hepatic Cytochrome P450 (CYP) Enzymes"

_pharmaceuticals, 2020, doi:10.3390/ph13090222_

Round 1

Reviewer 1 Report

The authors have modified the article extensively.

Although the authors did not respond to my previous major points precisely, they now included information on some other drug-metabolizing enzymes in Table1. They also changed the title into the one focusing on hepatic P450 enzymes, which seemed more suitable for this review.

Reviewer 2 Report

The manuscript has been revised according to the reviewers' comments.

Reviewer 3 Report

All questions were answered.

This manuscript is a resubmission of an earlier submission. The following is a list of the peer review reports and author responses from that submission.

Round 1

Reviewer 1 Report

Jamwal has very well summarized that impact of NAFLD on the drug- metabolizong protein expression is an important parameter controlling pharmacokinetic as well as the pharmacological effects of several drugs.

Thanks for this very interesting review.

Author Response

The author thanks the reviewer for their time and support of this review. 

Reviewer 2 Report

Nonalcoholic fatty liver disease (NAFLD) is a common disorder and characterized by excessive fats in the liver. Steatohepatitis (NASH) is a severe form of NAFLD which accompanied by inflammation, fibrosis, and hepatocyte damage in addition to steatosis. This review summarized the current knowledge of drug metabolism enzymes (DMEs) in NAFLD. The manuscript was well prepared and written. There are few comments as following:

  1. Please summarize the abbreviations used in the review.
  2. Please carefully proof read the manuscript to correct all the mistakes, like page 2, line 41, “ Nonalcoholic fatty liver (NAFL) liver….”
  3. The section numbers need to correct. Using “2.1”, “2.2”……
  4. Page 4, the summary of the DMEs was not clear. A figure or table will be helpful for the reader to follow.
  5. Page 5, a recently developed NASH model from Dr. Scott Friedman using a simple diet and chemical-induced mouse NASH model should be discussed. (PMID: 29572095)
  6. Page 9, hepatic blood flow although critical for NAFLD, but no connection to DMEs was discussed. In addition, most of those DMEs having zonation expression. The zonation of DME expressions should be discussed.

Author Response

The author thanks the reviewer for their time and support of this review. 

 All the reviewer comments have been addressed. Specifically, 

  1. Abbreviations have been added to the manuscript
  2. The manuscript has been proofread for errors in written English.
  3. As suggested, a table of the DMEs has been added (table 1)
  4. Information on recent murine models has been added on page 5.
  5. Metabolic zonation has been discussed in the manuscript (page 7).

Reviewer 3 Report

This is a review article focused on the important link between non-alcoholic fatty liver disease and the level of drug-metabolizing enzymes in the liver. The topic is of interest and some important aspects are covered in the paper. However, the manuscript should be improved.

  1. The structure of the paper is a little bit unbalanced. The section focused on general characteristics of NAFLD is relatively long whereas that about background of DME is quite short. It would be reasonable to expand this section and possibly to include the table presenting major DMEs (name, catalyzed reaction, etc.).
  2. In addition, the separate table summarizing the major features of most commonly used animal models of NAFLD could be included.
  3. The content of table 1, that is anti-NAFLD drugs under development, is not directly associated with the main topic of this paper. Authors could consider to remove this table.
  4. Lines 174-178: the section about drugs under development is not appropriate in this place.
  5. Do the authors obtained permission from the authors of original study to include the results presented in Table 3 in the paper?
  6. Title of table 3: activity levels of major P450 enzymes are not shown in the table.
  7. A brief section about possible implications of DME changes in NAFLD would be appreciated.
  8. A lot of language and grammar corrections are to be made. The manuscript should be revised by someone proficient in English writing.

Minor comments:

  1. Line 8: “excessive fats” should be corrected to “excessive fat:”
  2. Line 10: “accompanied with inflammation” should be corrected to “accompanied by inflammation”.
  3. Line 15: “pre-clinical species” should be corrected to “pre-clinical studies”. There are no “pre-clinical species”.
  4. Line 45, the phrase “liver hepatocytes” should be revised. There are no other hepatocytes.
  5. Lines 107-18: the sentence needs style revision.
  6. Line 109, “…diets rich in cholesterol, fructose and other saturated fats…”. Cholesterol and fructose are not saturated fats, so the sentence should sound: “cholesterol, fructose and saturated fat”.
  7. Line 141: :”…drugs in the market” should be corrected to: “…drugs on the market”.
  8. Line 158: “LDLR” is the abbreviation of LDL receptor, not of LDL receptor deficient mice.
  9. Line 183, “critical to understanding” should be corrected to: “critical to understand”.
  10. Line 188: “ADME” should be corrected to “DME”.
  11. Line 192: alterations of drug metabolizing enzymes in NAFLD are summarized in Table 2, not table 1.
  12. Line 204, legend for figure 2, “is increase” should be corrected to: “is increased”.
  13. Line 262, “there is a significant amount of literature is available”; the second “is” should be removed.
  14. Line 318: title of table 3 should be removed.

Author Response

The author thanks the reviewer for the valuable feedback on the manuscript. 

All the comments has been addressed in the revised version of the manuscript. 

The article structure has been modified.

Anti-NAFLD drugs under development table has been removed

The manuscript has been revised for language and grammatical corrections have been made.

Data for DMEs from animal models of NAFLD were not included as authors intend to restrict this review to available human data. 

Reviewer 4 Report

This review highlighted the potential significance of drug-metabolizing enzymes in the liver of NAFLD. Although the information on future drugs for NAFLD and the discussion about new proteomic platforms such as SWATH-MS are interesting, there is too little information on the topic itself: the only summarized data about the topic were shown in Table 2. There should be more to be discussed. For example, it is well known some enzymes such as AKRs or UGTs are significantly upregulated in NAFLD livers including humans, which should be mentioned and discussed in this context. Also, the potential interaction of future drugs in Table1 and drug-metabolizing enzymes would be interesting and clinically important. Thus, the authors missed so many important issues which are necessary for an overview.

Minor points

  • Some Gene Symbols should be designated in italics, but some appeared in orthographic letters.
  • There are some typos, such as PNLA3 (line62, which should be PNPLA3).

Author Response

The authors thank the reviewer for their time and comments. 

The manuscript has been revised for language and written English.